# JAKinhibs in Psoriatic Disease: Analysis of the Efficacy/Safety Profile in Daily Clinical Practice

**DOI:** 10.3390/diagnostics14100988

**Published:** 2024-05-08

**Authors:** Francesco Bizzarri, Ricardo Ruiz-Villaverde, Pilar Morales-Garrido, Jose Carlos Ruiz-Carrascosa, Marta Cebolla-Verdugo, Alvaro Prados-Carmona, Mar Rodriguez-Troncoso, Enrique Raya-Alvarez

**Affiliations:** 1Servicio de Reumatologia, Instituto Biosanitario de Granada, Ibs, Hospital Universitario San Cecilio, 18012 Granada, Spain; f.bizzarri95@gmail.com (F.B.); pimoraga@gmail.com (P.M.-G.); enriraya@gmail.com (E.R.-A.); 2Servicio de Dermatología, Instituto Biosanitario de Granada, Ibs, Hospital Universitario San Cecilio, 18012 Granada, Spain; ruizcarrascosa@movistar.es (J.C.R.-C.); martacevers@gmail.com (M.C.-V.); mar.troncoso10@gmail.com (M.R.-T.)

**Keywords:** JAK inhibitors, psoriasis, psoriatic arthritis

## Abstract

Psoriatic disease (PsD) affects multiple clinical domains and causes a significant inflammatory burden in patients, requiring comprehensive evaluation and treatment. In recent years, new molecules such as JAK inhibitors (JAKinhibs) have been developed. These have very clear advantages: they act quickly, have a beneficial effect on pain, are well tolerated and the administration route is oral. Despite all this, there is still little scientific evidence in daily clinical practice. This observational, retrospective, single-center study was carried out in patients diagnosed with PsA in the last two years, who started treatment with Tofacitinib or Upadacitinib due to failure of a DMARD. The data of 32 patients were analyzed, and the majority of them (75%) started treatment with Tofacitinib. Most had moderate arthritis activity and mild psoriasis involvement according to activity indices. Both Tofacitinib and Upadacitinib demonstrated significant efficacy, with rapid and statistically significant improvement in joint and skin activity indices, C-reactive protein reduction, and objective measures of disease activity such as the number of painful and inflamed joints. Although there was some difference in the baseline characteristics of the cohort, treatment responses were comparable or even superior to those in the pivotal clinical trials. In addition, there was a low frequency of mild adverse events leading to treatment discontinuation and no serious adverse events. These findings emphasize the strong efficacy and tolerability of JAKinhibs in daily clinical practice, supporting their role as effective therapeutic options for patients with PsD.

## 1. Introduction

Psoriatic disease (PsD) is a term that was first proposed in 2006 by Scarpa et al. [1] and describes a chronic, heterogeneous inflammatory disease that is composed of a wide variety of pathologies that share common immunological pathways, such as psoriasis (PsO), psoriatic arthritis (PsA), inflammatory bowel disease (IBD) and uveitis. Skin, nails, entheses, axial and peripheral joints are frequently affected, and the gastrointestinal tract and the eye are affected less frequently. In addition to these more “classical” clinical domains, recent research indicates that individuals diagnosed with PsD frequently exhibit an increased likelihood of cardiovascular complications, marked by a greater occurrence of accompanying conditions like diabetes, abnormal lipid levels, high blood pressure, obesity, metabolic syndrome, and various cardiovascular disorders such as heart attacks, strokes, and thromboembolic events [2].

### 1.1. Psoriasis

PsO impacts as many as 3% of the global populace with an increasing incidence over the years, has no clear gender predilection and has a significant psychological impact on patients. A study by Rapp et al. [3] shows that PsO, compared to a number of chronic diseases, such as myocardial infarction, cancer and congestive heart failure, is one of the most physically and psychologically debilitating diseases, considerably worsening the results of quality of life questionnaires. Psoriasis was surpassed in impacting psychological well-being only by depression and chronic lung disease [3].

There are five types of PsO:
-Plaque PsO, also known as “psoriasis vulgaris”, is the most common (approximately 90% of cases) and is characterized by well-defined, erythematous, scaly plaque lesions, usually concentrated on the extensor surfaces of the limbs, the periumbilical, perianal and retro auricular region and the scalp. Nail psoriasis results from the involvement of the nail bed or matrix and is quite common in patients with PsO (40–45%), significantly more prevalent in patients with PsA (80–90%), and is particularly associated with involvement of the distal interphalangeal joints. The clinical manifestations are variable, and alterations such as nail pitting, onycholysis, subungual hyperkeratosis, changes in nail plate coloring, and onychodystrophy can be observed. This functional and aesthetic alteration of the nails also entails an added psychological burden on the patient and is currently a therapeutic challenge due to the low response rate to both topical and systemic treatments [4,5].-Pustular PsO is characterized by white, coalescing, sterile pustules about 2–3 mm in size. Clinically, it is distinguished into two variants.-Generalized pustular PsO may develop in people with no history of PsO or occur in people with previous PsO vulgaris. It is characterized by dark, scattered erythematous plaques with sterile pustules, which coalesce to form purulent aggregates. Skin lesions may progress rapidly, and the disease is life-threatening. The skin lesions are associated with symptoms of systemic involvement such as high fever and malaise, together with elevated acute phase reactants and neutrophilic leukocytosis.-Localized forms of pustular PsO include palmoplantar pustular PsO and acrodermatitis continua suppurativa. Interestingly, tumor necrosis factor inhibitors (anti-TNF), which are effective therapies for treating PsO, have also been associated with the development of localized forms of pustular PsO [6,7].-Guttate PsO typically presents with an abrupt appearance of numerous erythematous papules about 2–6 mm in diameter, scaly and pruritic, in the form of a teardrop or droplet. It usually appears on the trunk and proximal extremities, while the palms and soles are generally unaffected. It is usually the most typical form of PsO in childhood. Sometimes, as a background, there may have been a previous streptococcal infection, mainly of the upper respiratory tract [6].-Inverse PsO, also known as flexural or intertriginous, is a variety of plaque PsO that affects the body folds, most commonly the axillary, anogenital and inframammary folds. It may occur alone or, more frequently, may be accompanied by plaque PsO in other locations. Bacterial or fungal superinfections (mainly Candida species) are common, as persistently moist skin provides an ideal environment for the growth of micro-organisms [6].-Erythrodermic PsO is a rare but very severe complication of psoriasis, occurring in 1–2% of patients. Clinically, it is characterized by the appearance of exuberant scaling erythema affecting at least 80–90% of the body surface. Due to extensive skin involvement, patients may present with systemic symptoms (fever with chills, intense pruritus, signs of dehydration, arthralgias, asthenia and lymphadenopathy). Several factors have been described as possible triggers, such as infections, systemic glucocorticoids, changes in medication, emotional stress, etc. [6].

### 1.2. Psoriatic Arthritis

PsA manifests as a persistent inflammatory condition characterized by diverse manifestations across various bodily tissues and clinical aspects such as arthritis, spondylitis, enthesitis, and dactylitis. Around 30% of individuals affected by PsO might eventually experience PsA, particularly those with severe PsO or noticeable nail or scalp symptoms. The estimated prevalence of PsA is between 30 and 100 cases per 10,000 people.

PsO typically comes before the onset of arthritis by around 10 years on average, yet in up to 15% of instances, arthritis might coincide with the beginning of skin issues or manifest even earlier [8]. Historically, five patterns of PsA have been described, which may change and overlap during the course of the disease: polyarticular, oligoarticular, distal, arthritis mutilans and axial or spondyloarthritis. The current trend is to differentiate three clinical forms: peripheral, axial and mixed [9].

Peripheral manifestations:

Peripheral manifestations include peripheral arthritis, dactylitis and enthesitis.

The distal subtype affects the interphalangeal joints of the hands and feet. Onychopathy is particularly common in this subtype of PsA. The oligoarticular form impacts four or fewer joints and typically manifests with an asymmetric distribution. Conversely, the polyarticular type affects five or more joints, often symmetrically resembling rheumatoid arthritis (RA). Mutilating arthritis, a deforming and destructive subtype that involves marked bone destruction and manifests with telescoping and unstable fingers, is the most severe and least common type of PsA, occurring in less than 5% of patients and causing significant deformity and disability. Radiographically, peripheral joints affected by PsA show eccentric erosions and narrowing of the joint interlining together with bony proliferation and periostitis, all occurring in the same joint, giving the typical “pencil-in-cup” figure [10].

Peripheral manifestations of PsA also include enthesitis and dactylitis.

The site where a tendon, ligament, or joint capsule connects to the bone is named enthesis, aiding in the movement of the joint. Enthesitis is inflammation of one of these structures. It can be found in up to 50% of patients and most commonly affects the plantar fascia and Achilles tendon but can also affect the patellar tendon, iliac crest, epicondyles and supraspinatus insertions [11].

Dactylitis, often referred to as “sausage toe”, is characterized by diffuse swelling of an entire finger or toe. It uniformly affects the soft tissues between the metacarpophalangeal and interphalangeal joints and is seen in 40–50% of patients. It is often seen in the context of polyarthritis, and in up to two-thirds of patients, it manifests only in the feet. Dactylitis may manifest as either acute symptoms (such as skin swelling, redness, and pain) or chronic symptoms (characterized by swelling without inflammation), frequently correlating with a more serious progression of the underlying disease [12].

Axial manifestations:

PsA exhibits similar genetic and clinical traits to various types of spondylarthritis (SpA), falling under this category of disease. The axial variant is distinguished by spinal and/or sacroiliac joint inflammation and subsequent structural alterations. Axial symptoms are present in 25–70% of PsA patients, with only a small fraction (2% to 5%) experiencing solely axial involvement; typically, this latter subgroup of patients also presents peripheral arthritis. A common feature of axial involvement is inflammatory rickets or low back pain characterized by insidious onset, morning stiffness > 30 min, improvement with exercise and worsening with rest, and nocturnal pain, especially in the second half of the night [13].

Factors contributing to the development of axial involvement in PsA include being positive for HLA-B27, having peripheral joint damage visible on radiological scans, and experiencing elevated erythrocyte sedimentation rate (ESR) [14].

While axial PsA typically presents with less severity compared to ankylosing spondylitis (AS), it notably exacerbates the disease’s impact on patients’ quality of life. The existence of the HLA-B27 genetic variation correlates with the heightened severity of PsA, and these genetic variations occur more commonly among patients experiencing axial involvement [15].

In certain instances, PsA may involve the axial skeleton without showing any clinical symptoms, with detection relying solely on radiographic or radiological examination. A prevalent characteristic of axial PsA is radiographic sacroiliitis, occurring in 25% to 50% of patients and often displaying asymmetry. The spine typically exhibits distinctive structural changes such as thick, asymmetrical non-marginal syndesmophytes. Additionally, early cervical spine involvement, including interfacial joint fusion, may occur independently of sacroiliitis or significant joint issues elsewhere in the spine.

Magnetic resonance imaging can help detect those inflammatory changes present in the early stages of the disease not identifiable in a radiographic study, such as bone and soft tissue edema in the sacroiliac joints and vertebral bodies [15].

### 1.3. Extra Musculoskeletal and Extracutaneous Manifestations

PsD is associated similarly to other SpA with a variety of immune-mediated extra-musculoskeletal manifestations, such as uveitis and IBD. The likelihood of experiencing these manifestations is greater among individuals with PsA in contrast to those solely with PsO or within the broader population [16].

In patients with axial PsA, IBD manifests in 11% of cases, showing a notably higher prevalence compared to individuals with peripheral-only PsA (2%) [13].

Symptoms that should lead to a preferential referral to a gastroenterologist are chronic diarrhea with bloody stools, chronic abdominal pain, symptoms that do not respect night rest, perianal fistula or abscess. In addition, other general signs and symptoms associated with these manifestations are oral thrush, anemia, fever and weight loss. However, it should be noted that PsD patients may often have subclinical forms, detectable only through high fecal calprotectin values and endoscopic studies [17].

Uveitis is the main ocular complication in patients with PsD, with 10% of them reporting ophthalmic complications at some point. Among PsD cases, anterior uveitis (AU) emerges as the predominant ocular presentation. Its prevalence varies between 2 and 25% and is predominantly observed in individuals with axial or HLA-B27-positive disease. Characteristic symptoms include severe light sensitivity, eye discomfort, excessive tearing, noticeable eye redness, and, in severe cases, blurred vision due to significant inflammation in the front chamber of the eye. Although recurrent UA is common, simultaneous involvement of both eyes can occur, which is rare in UA within the context of SpA [17,18,19].

It is good to keep these two common non-joint symptoms in mind when selecting the optimal treatment for patients.

### 1.4. Cardiovascular Comorbidities

In more recent studies, it has been demonstrated that individuals diagnosed with PsA experience a greater occurrence and frequency of cardiovascular disease (CVD) compared to the overall populace. This heightened susceptibility is not solely due to a higher occurrence of typical risk factors like high blood pressure, obesity, diabetes, and high cholesterol levels, but also because of persistent systemic inflammation. Increased carotid atheromatous plaques have been found in these patients, suggesting that exposure to chronic inflammation may influence the excess cardiovascular risk [20].

Recent evidence from observational studies demonstrates that patients with PsD are exposed to an increased risk of mortality compared to the general population, predominantly secondary to ischemic heart disease, stroke, and heart failure [21].

Among cardiovascular risk factors, hypertension has recently been found to be the most prevalent comorbidity in patients with PsA (39% of patients), followed by hyperlipidemia, diabetes and obesity [22]. Experts suggest that the systemic inflammatory state to which these patients are exposed can facilitate the increase in body weight and thus obesity. This can cause chronic stress to the joints, altering the physiological mechanics of joint function and generating injuries and microtrauma, the basis for the development of enthesitis and synovitis [23], and hinder response to treatment [17].

### 1.5. Other Comorbidities

In addition to the above, an increased incidence of hepatic steatosis, metabolic syndrome, depression, anxiety, fibromyalgia and osteoporosis has been reported in patients with PsD [17,24].

### 1.6. Assessment of Disease Activity

Assessing the degree of disease activity, both for the musculoskeletal domain and the severity and extent of PsO, is essential to choosing the most appropriate treatment and being able to apply modern “treat to target” treatment strategies.

To evaluate the degree of skin involvement, there are several tools. The Psoriasis Area Severity Index (PASI) is the most widely used score to assess the severity of psoriasis, and the PASI75 (75% or greater reduction in PASI scores from baseline) represents one of the most widely used targets in psoriasis trials. It is a combination of the severity of skin involvement with the extent of the affected area in a single score ranging from 0 (no disease) to 72 (maximum disease). The body is divided into four areas (head, upper limbs, trunk and lower limbs), and for each area, the percentage (0% = 0 points; 100% = 6 points) of affected skin and the severity of erythema, induration and scaling (0–4 points) are estimated. Each of these areas has a different weight in the final calculation and scores on its own; once the degree of involvement has been calculated for each of these, they are combined to obtain the final result [25,26].

When joint involvement is present, clinical trials usually measure the degree of activity using rheumatoid arthritis-derived tools such as the American College of Rheumatology (ACR) 20, ACR 50 and ACR 70 response rate (indicating reductions in the number of painful and swollen joints) of at least 20%, 50% and 70% from baseline, together with an improvement in at least three of the following five items: a global assessment of arthritis by the patient and the responsible physician measured on a visual analog scale (VAS), degree of pain by the patient measured on a VAS, degree of disability measured by the HAQ-DI, and changes in C-reactive protein (CRP) levels [27].

There are several composite scores that measure disease activity taking into consideration various domains, such as axial and peripheral skin and joint involvement [28]; among them, one of the most widely used in clinical practice is the Disease Activity Index for Psoriatic Arthritis (DAPSA) and is based on the sum of five variables: tender and swollen joint count, global assessment of the patient’s disease on a 10 cm VAS, as well as global assessment of the patient’s pain on the same scale and CRP as an acute phase parameter [29]. It has been validated for use in PsA, where it has been shown to correlate well with disease activity and ultrasound-assessed synovitis [30].

### 1.7. Aetiopathogenesis

PsD shows a complex and not yet fully elucidated pathophysiology. The significant role of the major histocompatibility complex as a pivotal susceptibility site for PsA and PsO is commonly acknowledged, evident in the statistic that roughly a quarter of PsA patients exhibit positivity for human leukocyte antigen HLA-B27 [31].

The key feature of PsO is the uncontrolled proliferation of keratinocytes, secondary to proinflammatory stimuli sustained by the activation of multiple cellular pathways. Activation of plasmacytoid dendritic cells by stimuli derived from damaged keratinocytes and interferon-α production activates myeloid dendritic cells, which in turn promote the activation and differentiation of T helper type 1 (Th1) and T helper type 17 (Th17) cells in the lymph nodes. These lymphocytes return to the dermis where they release interleukin-12, 17 and 22 and tumor necrosis factor α (TNF-α), along with a variety of chemokines and other cytokines [32]. One of the most important discoveries regarding the etiopathogenesis of PsD has been the role of the IL-23/Th17 axis in promoting the inflammatory process. This axis, in addition to promoting keratinocyte proliferation and other features typical of psoriasis, has been found to be closely related to the pathogenesis of musculoskeletal manifestations. Thus, in entheses, the release of IL-23 in response to biomechanical stress or microtrauma activates TH17 and the production of IL-22 and TNF-α with consequent inflammation, bone erosion and pathological bone formation.

In response to these cytokines, mesenchymal cells differentiate into osteoblasts, forming enthesophytes and syndesmophytes in the spine. Increased expression of NF-κB receptor activator ligand (RANKL) by synoviocytes, together with the pro-inflammatory cytochemical environment, drives the differentiation of precursors into osteoclasts, with subsequent production of synovitis and bone resorption [10].

Many of these aberrantly produced proinflammatory cytokines and growth factors act as ligands for receptors attached to intracytoplasmic tyrosine kinases called JAKs. There are four types of JAK: JAK 1, JAK 2, JAK 3 and TYK 2. These tyrosine kinases are able to phosphorylate tyrosine residues of other adjacent molecules called STAT. The latter is a family of transcription factors that act by downstream signaling to JAKs and consist of seven members (STAT1, STAT2, STAT3, STAT4, STAT5a, STAT 5b and STAT6) [33].

Thus, JAKs activate STAT proteins, which move to the nucleus, thereby modulating the synthesis of proinflammatory nuclear factors. The JAK/STAT pathway mediates the effect of many different molecules in the body such as growth and colony-stimulating factors, hormones, cytokines, interleukins and interferons. Among all these cytokines, IL-23 and type I interferons, which play a key role in the pathogenesis of PsD, bind to their transmembrane receptor which transduces the signal through JAK-type kinases.

### 1.8. JAKinhibs in Psoriasic Disease

The increasing understanding of the etiopathogenic mechanisms of these immune-mediated diseases and of the proinflammatory intracellular signaling pathways involved in PsD have made it easier to discover and create novel small molecules capable of inhibiting JAK proteins, emerging as effective treatment options for rheumatic and dermatological manifestations [2].

While biological disease-modifying antirheumatic drugs (DMARD) (DMARDb) are monoclonal antibodies directed against one or a small subset of cytokines, such as inhibitors of IL-17, IL-23, IL-12/23 or TNF-α, JAK inhibitors (JAKinhibs) simultaneously suppress the activation and production of multiple cytokines involved in the Th1, Th2, Th17 and Th22 immune pathways.

In addition, JAKinhibs are a class of drugs that act rapidly, within a few days, with a striking beneficial class effect on pain.

The oral formulation they share makes them a well-tolerated class of drugs, with a clear advantage over DMARDb in terms of convenience of administration, improving therapeutic compliance.

Another aspect to highlight is that these small molecules do not suffer from immunogenicity problems.

To date, the EMA/FDA has approved Tofacitinib and Upadacitinib for the treatment of PsA and Deucravacitinib for PsO.

### 1.9. Tofacitinib

Tofacitinib is a major inhibitor of JAK1 and JAK3 and minor inhibitors of JAK2 and TYK2 approved by the FDA and EMA for the treatment of active PsA in combination with MTX in patients with inadequate response or intolerance to a previous DMARD. The efficacy and safety of Tofacitinib were evaluated in two randomized, double-blind, placebo-controlled Phase III trials (OPAL BROADEN and OPAL BEYOND) [34,35] that were performed in adult patients with active PsA, defined as a tender and swollen joint count equal to or greater than three, and active plaque PsO. Patients included in the OPAL BROADEN study were anti-TNF naïve and an active comparator (Adalimumab 40 mg/2 weeks) was also included in the study. For efficacy outcomes, the ACR20 response and the Health Assessment Questionnaire Disability Index (HAQ-DI) at 3 months from the start of treatment were selected as primary endpoints. Secondary variables included, among others, the ACR50 and 75 response and PASI75.

Detailed results are shown in Table 1.

In summary, treatment with Tofacitinib resulted in significantly higher ACR20 response rates compared to placebo at month 3 in both naïve and inadequate anti-TNF responders. In addition, significant differences were observed from week 2, demonstrating the rapidity of action of the drug. The HAQ-DI response rate at month 3 in both studies was statistically significantly higher in patients receiving Tofacitinib compared to those receiving placebo.

Results for secondary endpoints were also consistent (Table 1).

Patients who participated in the pivotal trials were enrolled in an open-label extension study with a long-term follow-up of up to 48 months (OPAL BALANCE) [36]. This study included a total of 685 patients and found that improvements in joint and skin outcomes were sustained over time. It also provided important long-term safety data, reporting that in patients treated with Tofacitinib, the most common adverse events were mild upper respiratory tract infections, headache, and gastrointestinal symptoms (diarrhea, nausea, vomiting) at a rate comparable to placebo and Adalimumab. The rate of treatment discontinuation due to adverse effects in the first 3 months was comparable (around 2%).

The incidence rate of serious infections (pneumonia most frequently) in patients treated with Tofacitinib was slightly higher than with Adalimumab, although this did not reach statistical significance.

Regarding Herpes Zoster (HZ) infection, the number of cases of reactivation was higher in patients treated with Tofacitinib than in the other two groups; however, the majority of HZ cases resolved without complications.

Dose-dependent cytopenias (anemia, leukopenia, lymphopenia) and elevation of liver cytolysis enzymes and lipid parameters (total cholesterol, LDL, HDL) were reported.

Overall, the safety profile observed in patients with PsA is consistent with the profile of rheumatoid arthritis patients, in terms of the type and frequency of adverse effects observed, except for major cardiovascular events, where long-term data demonstrated a lower incidence in the PsA patient cohorts compared to the results of rheumatoid arthritis studies [2].

### 1.10. Upadacitinib

The efficacy and safety of Upadacitinib, a specific blocker of JAK1 and, to a minor degree, JAK2, have been assessed in two pivotal 24-week double-blind phase III clinical trials (CTs) (SELECT-PsA-1 and SELECT-PsA-2), and it is currently endorsed by both the EMA and FDA for managing PsA [37,38].

The SELECT-PsA-1 trial comparing Upadacitinib at different doses (15 mg or 30 mg once daily) with Adalimumab 40 mg every 2 weeks subcutaneously and placebo (1:1:1:1) in approximately 1705 adult patients with active PsA and inadequate response to at least one synthetic DMARD (DMARDs) found that ACR20 response rates (primary response endpoint) at week 12 were significantly lower than those of placebo: (1) in approximately 1705 adult patients with active PsA and inadequate response to at least one DMARDs found that the ACR20 (primary response variable) response rates at week 12 of the groups treated with the new molecule were higher than the rates of placebo patients and not lower than the rates of Adalimumab (Table 1).

This study also demonstrated the efficacy of Upadacitinib on skin outcomes, as evidenced by good PASI75 response rates [37].

In SELECT-PsA-2, 642 patients with PsA and an inadequate response or intolerance to at least one DMARDb were randomized (2:2:1:1) to Upadacitinib 15 mg or 30 mg once daily, and two placebo groups. At week 12, ACR20 response rates of patients receiving Upadacitinib were significantly higher compared to placebo, as were skin outcomes.

Efficacy manifested rapidly in all responses, with the most pronounced ACR20 responses detected as early as week 2.

Secondary endpoints in both studies included, among others, ACR50 and ACR70 response rates at week 12 and week 16 (Table 1) [38].

In the trial extension periods of 260 weeks (5 years) in SELECT-PsA-1 and 156 weeks (3 years) in SELECT-PsA-2, at week 24, the placebo group‘s subjects were re-randomised to receive Upadacitinib. For these subjects, the improvements achieved after Upadacitinib initiation were similar to those observed in subjects who started Upadacitinib on day 1, with comparable results at week 56.

In terms of safety, data from Phase III RCTs show an acceptable safety profile, which is also comparable to that of Adalimumab. The primary adverse effects observed with Upadacitinib 15 mg were mild upper respiratory tract infections, nasopharyngitis, urinary tract infections, headaches, nausea and diarrhea, elevated creatine phosphokinase (CPK) and alanine aminotransferase blood transaminases and hypertension. Rates of serious infections (pneumonia most frequently), HZ, anemia, neutropenia and lymphopenia remained numerically higher with Upadacitinib 15 mg than with Adalimumab, but without statistically significant differences. Comparing all groups, it was seen that the rates of neoplasia (excluding non-melanoma cancer), venous thromboembolic disease, and adverse cardiovascular events were similar between them. Higher rates of serious infections and elevated liver transaminases have also been observed in patients treated with Upadacitinib in combination with MTX compared to patients treated with Upadacitinib alone. Overall, the safety profile of Upadacitinib in PsA was consistent with previous experience in patients with rheumatoid arthritis.

### 1.11. Deucravacitinib

In PsO, Deucravacitinib, a highly selective, allosteric oral TYK2 inhibitor, has recently been approved by the EMA and FDA for the treatment of moderate-to-severe plaque PsO based on data from its two phase III CTs POETYK PSO-1 and POETYK PSO-2 [39,40] (Table 1).

This molecule is also being studied for PsA (NCT03881059), and a recent phase II RCT, in which 203 patients with PsA were randomized to Deucravacitinib at doses of 6 mg and 12 mg once daily and placebo, has reported promising results, showing statistical significance in terms of differences in ACR 20 response rates at week 16 for both treatment groups compared to placebo [41].

Phase III RCTs are currently under development.

Inhibiting JAK kinases seems to hold potential as a promising therapy for PsD, and various RCTs are currently underway for alternative JAK inhibitors.

It is well known that in daily clinical practice, patients can differ greatly from those included in RCTs, in terms of demographic and clinical characteristics and response to treatments. That is why it is essential to look for real-world evidence by carrying out observational studies—this being extremely necessary to verify the data and results obtained in literature and improve daily decisional strategies.

### 1.12. Objectives

Therefore, the aim of our study is to evaluate the efficacy and safety of Tofacitinib and Upadacitinib (the only JAKinhibs currently approved for the treatment of PsA) in both the musculoskeletal and cutaneous domains of PsD in daily clinical practice, analyzing data from a cohort of patients seen in a shared multidisciplinary dermatology–rheumatology practice.

## 2. Materials and Methods

This was an observational, retrospective, single-center study in which data were collected on patients diagnosed with PsA seen in the last 2 years (from December 2021 to December 2023) in the shared dermatology–rheumatology consultation of a second-level hospital in the province of Granada (Spain).

Only data from patients over 18 years of age who had been diagnosed with PsA according to the criteria of the Classification Criteria for Psoriatic Arthritis (CASPAR) group were included [42], who started treatment with Tofacitinib or Upadacitinib due to failure of one or more DMARDb and who had no major contraindications to starting treatment with that class of drugs (>65 years, personal history of cardiovascular events, multiple cardiovascular risk factors, personal history of malignancy, high risk of thromboembolic disease, pregnancy and/or breastfeeding, active infection or personal history of M. tuberculosis, hepatitis B or C virus infection). All patients were refractory to at least 1 DMARDb. Tofacitinib was used at a dose of 5 mg orally twice daily or at a dose of 11 mg once daily and Upadacitinib at a dose of 15 mg once daily.

Follow-up at our clinic was carried out with quarterly appointments, and patients attended once a month for control and monitoring tests at their health center, where they were seen by a primary care physician.

Every continuous variable underwent Shapiro–Wilk normality testing, and outcomes were presented either as mean ± standard deviation (SD) or as median and interquartile range [IQR], depending on suitability.

Variables collected at the baseline visit included age, sex, concomitant diseases, time of disease progression, presence of radiographic erosions, number of previous biologic treatments, concomitant treatment with any DMARDs (Methotrexate, Leflunomide or Sulphasalazine), chronic glucocorticoid treatment and dose.

The outcome variables were, at the baseline appointment and at each subsequent appointment, the number of painful joints (PJC), number of swollen or inflamed joints (SJC), patient’s global assessment of arthritis activity measured through a VAS ranging from 0 to 100 mm, patient’s global assessment of pain secondary to arthritis activity measured through a VAS ranging from 0 to 100 mm, patient’s global assessment of pain secondary to arthritis measured by VAS ranging from 0 to 100 mm, assessor’s global assessment of arthritis measured by VAS ranging from 0 to 100 mm, CRP (mg/L), erythrocyte sedimentation rate (ESR), PASI and DAPSA.

At 3 months after initiation of treatment, ACR 20/50/70 response and PASI75 response rates were calculated for each patient. To find differences, outcome variables were assessed and compared between the baseline and 3-month visit using paired t-tests or Wilcoxon signed-rank tests depending on the normality distribution.

Statistical significance was established with a value of *p* < 0.05, and the analyses were carried out through the R program, version 4.1.2 (R Foundation for Statistical Computing).

This study was not funded by any pharmaceutical company. It was the result of an independent initiative by the investigators, was conducted according to the guidelines of the Declaration of Helsinki, and approved by the Provincial Ethics Committee of Hospital Services of Granada with the code HUSC_DER_REU_2004_001.

## 3. Results

### 3.1. Baseline Demographic and Clinical Characteristics

Data from 32 patients (24 women and 8 men) were included, of whom 24 (75%) started treatment with Tofacitinib and 8 (25%) with Upadacitinib (Table 2).

The mean age of the total patients was 45.25 ± 10.71 years, with some differences between the Tofacitinib (47.38 ± 9.55) and Upadacitinib (38.88 ± 12.10) groups.

Each patient included in the study met the CASPAR classification criteria for the diagnosis of PsA. The pattern of joint involvement was peripheral (*n* = 26), mixed (*n* = 6) and without pure axial PsA. The mean time of disease progression from diagnosis of PsA to initiation of treatment with one of the two JAKinhibs was 6.5 ± 5.22 years, slightly longer in patients treated with Upadacitinib than in those who started Tofacitinib (Table 2).

Regarding clinical characteristics, of the total patients, 26 (81.25%) suffered from skin psoriasis at the time of starting JAKinhib.

In terms of radiographic features, six patients (18.75%) had objectionable erosions on hand radiographs at the time of treatment initiation.

Prior to the start of JAKinhib, all patients had received at least one DMARDb, with a mean of 3.53 ± 2.12 previous treatments received per patient.

At the time of initiation of Tofacitinib or Upadacitinib, seven patients (21.87%) were on concomitant treatment with a DMARD, four of which with methotrexate, one with sulphasalazine and one with leflunomide. In the remaining 25 (78.13%) patients, Jakinihb was used as monotherapy. The number of subjects treated with a concomitant corticosteroid regimen at the start of JAKinhib was 10 patients (31.25%), all on Prednisone 5 mg/day or lower.

Regarding analytical data, the median CRP levels were 4.70 (1.48–12.9) and 19.5 (7–28.5) for ESR before starting Tofacitinib or Upadacitinib.

As for baseline disease activity data, the median of the number of PJC and SJC was five (3–7.25) and three (2–4), respectively; the mean VAS of pain and activity according to the patient was 73.44 ± 14.28 and 70.94 ± 15.73, respectively; the mean VAS from the physician’s point of view was 64.38 ± 17.03.

As for the composite activity indices, the mean baseline DAPSA was 24.56 ± 7.97, with 2 patients (6.25%) classified as low activity, 22 (68.75%) as moderate activity and 8 (25%) as high activity; the median PASI of the 26 patients (81.25%) with psoriasis at baseline was 3.7 (2.1–5.3), with 21 patients (80.77%) with mild involvement, 5 (19.23%) with moderate involvement and none with severe involvement, according to the established thresholds. All five patients who received Upadacitinib and who had psoriasis at baseline had mild involvement based on PASI categories. The remaining baseline data broken down by the treatment subgroup are found in Table 2.

### 3.2. Efficacy Results

Overall, after the first 3 months of therapy with Tofacitinib and Upadacitinib, most patients experienced rapid joint improvement as well as good results in skin clearance. Most of the selected outcome variables (PJC, SJC, CRP, PASI, DAPSA) showed statistically significant changes as well as the response rates ACR20, ACR50, ACR70 and PASI7, which showed a significant improvement in the first quarter of therapy (Table 3).

The mean DAPSA at 3 months was 13.54 ± 8.21 with a reduction of 11.02 points compared to the baseline value and with a statistically significant result (*p* < 0.001) (Figure 1), achieving remission in 1 patient (3.33%), low disease activity in 17 patients (56.66%), with 10 patients (33.3%) remaining at moderate activity and 2 (6.66%) at high (Table 3). In the Tofacitinib group, the mean post-treatment DAPSA was 13.10 ± 8.54, with a statistically significant reduction (*p* < 0.001), and of the 22 patients who remained on treatment at 3 months (initial total of 24 patients, of whom 2 stopped treatment before 3 months), 14 patients (63.64%) achieved remission or low disease activity [1 patient (4.54%) achieved remission and 13 (59.10%) achieved low disease activity] (Figure 2 and Table 3). In the eight patients who were on treatment with Upadacitinib, the mean DAPSA at three months was 14.76 ± 7.63 (*p* = 0.003), and 50% of them achieved low disease activity; the other half remained at moderate activity (Figure 3 and Table 3). In terms of PASI, there was a reduction of almost three points after 3 months of treatment (Figure 4), with a median, referring to the total number of patients, of 0.8 [0–2.23], achieving statistical significance (*p* < 0.001). All patients were classified as “mildly affected” according to established thresholds (Table 3). In the Tofacitinb group, the mean PASI after treatment was 1.58 ± 1.55 (*p* < 0.001; Figure 5) and in the Upadacitinib group, the median was 0 [0–0.8] (*p* = 0.03; Figure 6).

Furthermore, after 3 months of treatment, statistically significant differences were detected in the reduction in CRP values (*p* = 0.02), TJC (*p* < 0.001) and SJC (*p* < 0.001). Likewise, ESR showed a trend towards reduction during follow-up, although without achieving statistical significance (*p* = 0.26). For the remaining data related to each of the drugs used, see Table 3.

At 3 months, ACR20 response rates with both Tofacitinib and Upadacitinib were significant, with 20 patients (66.66%) of 30 total achieving this outcome [14 (63.63%) with Tofacitinib and 6 (75%) with Upadacitinib]. Similarly good results were also obtained with ACR50 and ACR70 response rates of 50% and 26.66%, respectively (Table 3).

At the cutaneous level, both JAKinhibs proved effective, achieving PASI75 response rates of 58.33% (52.63% in the Tofacitinib group and 80% in the Upadacitinib group).

None of the patients who had no skin involvement at the time of initiation of JAKinhib therapy had any psoriatic lesions at 3 months.

The remaining data on outcome variables are summarized in Table 3.

### 3.3. Safety, Retention and Adverse Effects Data

The retention rate of JAKinhibs at 3 months was 93.75%, respectively, 91.66% in the Tofacitinib group and 100% in the Upadacitinib group. Two patients discontinued treatment with Tofacitinib before 3 months due to mild adverse events; the first patient discontinued treatment due to gastro-oesophageal reflux resistant to proton pump inhibitor treatment that ceased when Tofacitinib was withdrawn and the second due to diarrhea and a fixed drug exanthema skin reaction that was diagnosed by the dermatology department and subsided when treatment was discontinued. No serious adverse events were observed.

The patients had no mild or severe infections. There was no case of HZ reactivation.

No thrombotic or cardiovascular events were observed, no cytopenias or thrombopenias occurred, and mean hemoglobin, lipid and transaminase levels remained stable throughout the follow-up.

No neoplasm appeared during the treatment period.

## 4. Discussion

JAKinhibs represent a novel and interesting class of drugs that have been shown to be safe, effective and rapid in the treatment of PsD by inhibiting the action of several inflammatory cytokines important for the disease process.

As mentioned above, both Tofacitinib and Upadacitinib have demonstrated efficacy in RCTs of PsA refractory to DMARDs (OPAL Broaden and SELECT-PsA-1) and TNF inhibitors (OPAL Beyond and SELECT-PsA-2). In the OPAL Beyond trial, which included patients refractory to at least one biologic therapy, as in our cohort, the 3-month ACR20 and 50 response rates with Tofacitinib at 5 mg/12 h were significantly higher compared to placebo, as were changes in HAQ-DI score. However, in terms of PASI75 response, the 5 mg/12 h dose failed to demonstrate superiority to the placebo. When analyzing the pooled data from the two phase III RCTs in the OPAL Balance posthoc study, statistically significant differences were found in favor of treatment with Tofacitinib compared to placebo in terms of the PASI75 response variable at 3 months [36].

As for Upadacitinib, in SELECT-PsA-2 in patients who had previously failed a DMARDb, the ACR20, 50, 70 and PASI75 response rates were significantly higher than the placebo, and long-term persistence was demonstrated.

Comparing the baseline characteristics of the patients in our cohort with those included in the pivotal RCTs [35,38], we can state that, on average, our patients were younger, with a greater prevalence of women and with a shorter duration of disease.

The TJC/SJC and the PASI scores were higher in patients in the trials.

In terms of treatment, in our cohort, there was a higher proportion of patients on low-dose corticosteroids at the start of the JAKinhib and that had received a greater number of prior DMARDs, while the proportion of patients on concomitant treatment with any DMARDs was higher in patients in the trials.

In terms of the results obtained, as in the trials of both treatments, our patients achieved a rapid and statistically significant improvement in joint activity (DAPSA) and skin activity (PASI) indices, with ACR20/50/70 and PASI75 response rates even higher than those obtained in the pivotal studies. Furthermore, interesting results were found regarding the substantial reduction in blood inflammatory parameters such as CRP and ESR, along with the significant reduction in objective measures of disease activity such as TJC and SJC. It should be noted that, at the time treatment when JAKinhib was initiated, our patients, on average, had less joint and skin involvement compared to those in the trials, and were also refractory to more biologics and had a lower rate of concomitant treatment with DMARDs. Therefore, the results obtained in our cohort reinforce the idea that both Tofacitinib and Upadacitinib have a very good efficacy profile in patients with psoriatic arthritis.

In terms of tolerability and safety, these drugs have proven to be very well tolerated and safe, with only 6.25% of patients experiencing mild adverse events leading to discontinuation of treatment. Significantly, we noted a decreased occurrence of negative outcomes in our investigation when contrasted with ECRs. The utilization of JAKinhibs as sole therapy in over three-quarters of our patient cohort, coupled with an average age of around 45 years, representing a younger demographic with minimal comorbidities, likely elucidates the diminished incidence of adverse events in our clinical setting.

### Limitations

Our study has several limitations. Firstly, the retrospective design of the study predisposes to a greater risk of bias; secondly, the small number of patients included, especially in the Upadacitinib treatment group, limits the power of the statistical tests performed. In addition, the follow-up period was short, with data only available at 3 months. Furthermore, it is a study that lacks comparison arms with other treatments, limiting itself to analyzing the responses to the two molecules used. Another limitation of this study was the lack of outcome variables studying responses in enthesitis and dactylitis. Lastly, regarding the high retention rates, we are aware that they could be artificially higher given that these are patients with refractory disease who have fewer therapeutic options.

## 5. Conclusions

As in the pivotal trials, our patients experienced a rapid, sustained and statistically significant improvement in the number of painful/inflamed joints, CRP and joint activity indices. A significant improvement in the PASI score was also observed. There were no major safety issues, as no patient experienced a serious adverse effect, and only two patients discontinued treatment due to minor adverse events.

In conclusion, our data support that Tofacitinib and Upadacitinib are effective, rapid and safe in daily clinical practice for the treatment of PsD refractory to multiple biologic therapies, despite clinical differences with patients included in the pivotal trials.

## Figures and Tables

**Figure 1 diagnostics-14-00988-f001:**
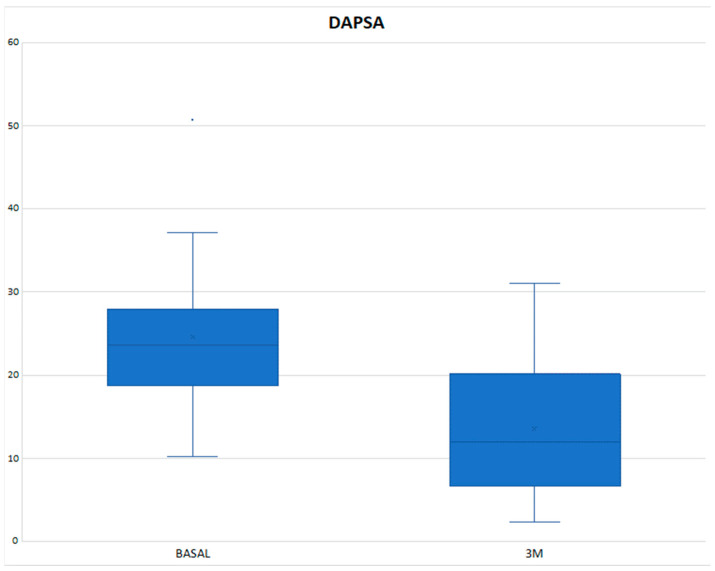
Differences in PsA activity scores measured with the DAPSA between baseline and 3 months of treatment in 32 patients who received a JAKinihb. DAPSA (Disease Activity Index for Psoriatic Arthritis); 3M (three months); * *p* < 0.05 vs. baseline.

**Figure 2 diagnostics-14-00988-f002:**
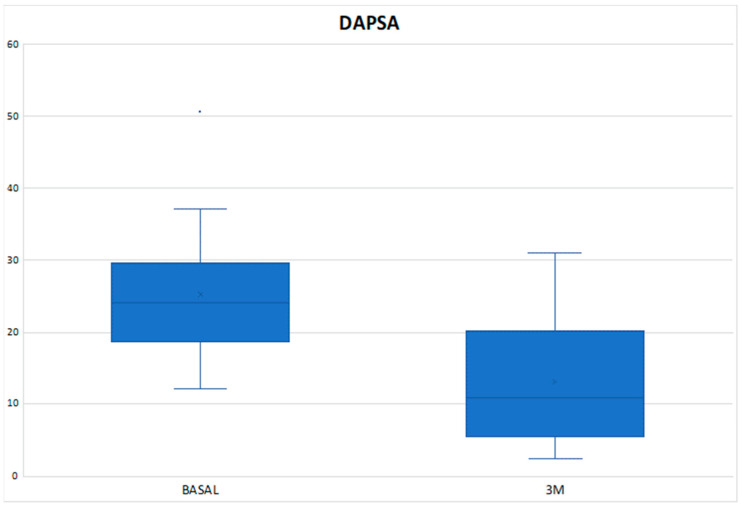
Differences in PsA activity score measured by DAPSA between baseline and 3 months of treatment in the 24 patients who received Tofacitinib. DAPSA (Disease Activity Index for Psoriatic Arthritis); 3M (three months); * *p* < 0.05 vs. baseline.

**Figure 3 diagnostics-14-00988-f003:**
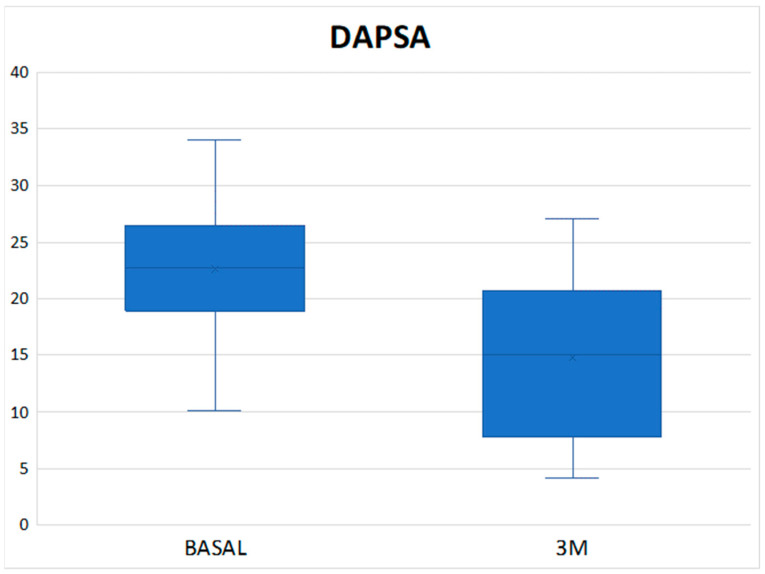
Differences in PsA activity score measured by DAPSA between baseline and 3 months of treatment in the 8 patients receiving Upadacitinib. DAPSA (Disease Activity Index for Psoriatic Arthritis); 3M (three months).

**Figure 4 diagnostics-14-00988-f004:**
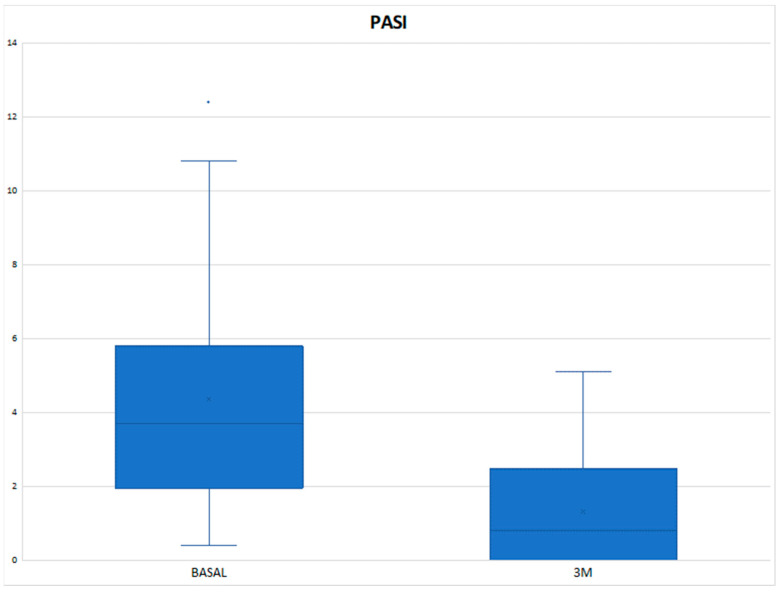
Differences in PsO activity score measured by PASI between baseline and 3 months of treatment in the 26 patients with psoriasis at baseline who received a JAKinhib. PASI (Psoriasis Area and Severity Index); 3M (three months); * *p* < 0.05 vs. baseline.

**Figure 5 diagnostics-14-00988-f005:**
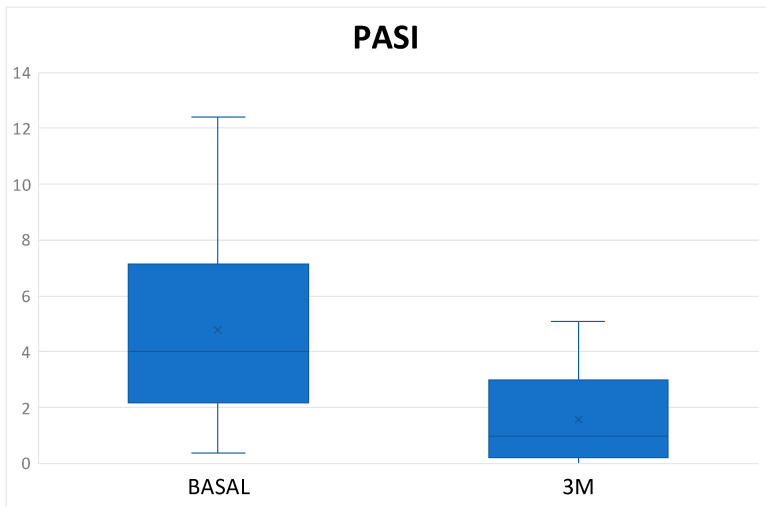
Differences in PsO activity score measured by PASI between baseline and 3 months of treatment in the 5 patients with psoriasis at baseline who received Upadacitinib. PASI (Psoriasis Area and Severity Index); 3M (three months).

**Figure 6 diagnostics-14-00988-f006:**
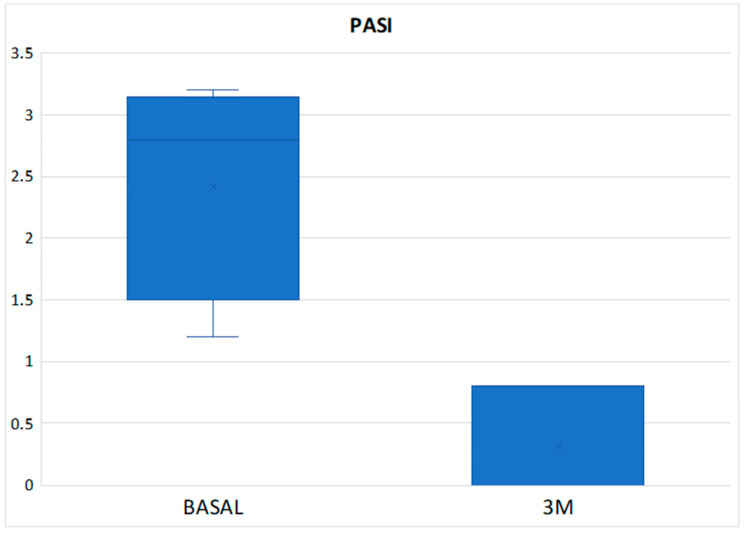
Differences in PsO activity score measured by PASI between baseline and 3 months of treatment in the 5 patients with psoriasis at baseline who received Upadacitinib. PASI (Psoriasis Area and Severity Index); 3M (three months).

**Table 1 diagnostics-14-00988-t001:** Comparative results of phase III RCTs of JAKinhibs in psoriatic disease.

Variable	TOFACITINIB					UPADACITINIB					DEUCRAVACITINIB					
	*OPAL BROADEN*			*OPAL BEYOND*		*SELECT PSA*-1			*SELECT PSA*-2		*POETYK PSO1*			*POETYK PSO2*		
	TOFA	ADA	PBO	TOFA	PBO	UPA	ADA	PBO	UPA	PBO	DEUCRA	APR	PBO	DEUCRA	APR	PBO
**ARC 20 ^a^**																
Week 12	50%	52%	33%	50%	24%	70.6%	65%	33.2%	56.9%	24.1%						
**ARC 50 ^a^**																
Week 12	28%	33%	10%	30%	15%	37.5%	37.5%	13.2%	31.8%	4.7%						
**ARC 70 ^a^**																
Week 12	17%	19%	5%	17%	10%	15.6%	13.8%	2.4%	8.5%	0.5%						
**PASI 75 ^b^**																
Week 12	43%	39%	15%	21%	21%											
Week 16						62.6%	53.1%	21.3%	52.3%	16%	58.4%	31.5%	12.7%	53%	39.8%	9.4%
**PASI 90 ^b^**																
Week 16											35.5%	19.6%	4.2%	27%	18.1%	2.7%
**PASI 100 ^b^**																
Week 16											14.2%	3%	0.6%	10.2%	4.3%	1.2%

^a^ ACR20/50/70, according to the American College of Rheumatology criteria, is defined as a 20/50/70% reduction since the start of treatment in the number of painful joints (out of 68 joints assessed) and swollen joints (out of 66 joints assessed) and a 20/50/70% improvement in at least three of the following measures: patient and physician global assessment of arthritis (measured on a VAS), patient assessment of arthritis pain (measured on a VAS), disability (measured by HAQ-DI) or CRP level. ACR20/50/70 response rates were calculated in a total of 30 patients as two patients discontinued Tofacitinib treatment before 3 months due to mild adverse events. ^b^ PASI75/90/100: 75/90/100% reduction in PASI score from the start of treatment. ADA: Adalimumab at a dose of 40 mg/2 weeks sc; APR: Apremilast 30 mg/12 h orally; DEUCRA: Deucravacitinib 6 mg/day orally; PBO: placebo; TOFA: Tofacitinib 5 mg/12 h orally; UPA: Upadacitinib 15 mg/day orally.

**Table 2 diagnostics-14-00988-t002:** Baseline characteristics of 32 patients with DMARDb-refractory PsA in clinical practice who started treatment with Upadacitinib or Tofacitinib.

	Total Patients (*n* = 32)	UPADACITINIB (*n* = 8)	TOFACITINIB (*n* = 24)
**Baseline demographics**			
Age, yrs	45.25 ± 10.71	38.88 ± 12.10	47.38 ± 9.55
Sex, M/F, *n* (%)	8/24 (25/75)	1/8 (12.5/87.5)	7/17 (29.17/70.83)
**Disease characteristics**			
PsA duration, yrs	7.88 ± 5.22	11.25 ± 7.89	
Radiographic lesions, *n* (%)	6 (20.63)	0	6 (25)
Prior biological therapies	3.53 ± 2.12	4.5 ± 2.93	3.21 ± 1.74
Concomitant csDMARD, *n* (%)	7 (21.87)	2 (25)	5 (20.83)
Methotrexate	1 (3.13)	1 (12.5)	0
Leflunomide	5 (15.63)	1 (12.5)	4 (16.67)
Sulfasalazine	1 (3.13)	0	1 (4.17)
Concomitant glucocorticoids, *n* (%)	10 (31.25)	4 (50)	6 (25)
Number of tender joints	5 [3–7.25]	4 ± 2.45	6.08 ± 3.59
Number of swollen joints	3 [2–4]	2.38 ± 1.06	3 [2–4.25]
VAS pain	73.44 ± 14.28	71.25 ± 16.42	74.17 ± 13.81
VAS activity	70.94 ± 15.73	67.50 ± 17.53	72.08 ± 15.32
VAS doctor	64.38 ± 17.03	57.50 ± 16.69	66.67 ± 16.85
CRP, mg/L	4.7 [1.48–12.9]	9.3 [2.33–24.7]	4.3 [1.3–9.7]
ESR, mm/h	19.5 [7–28.5]	23 ± 13.09	20 [5–28.5]
PASI ^a^	3.7 [2.1–5.3]	2.42 ± 0.88	4.8 ± 3.23
Mild, *n* (%)	21 (80.77)	5 (100)	16 (76.19)
Moderate, *n* (%)	5 (19.23)	0	5 (23.81)
Severe, *n* (%)	0	0	0
DAPSA ^b^	24.56 ± 7.97	22.57 ± 6.88	25.23 ± 8.32
Remission, *n* (%)	0	0	0
Low, *n* (%)	2 (6.25)	1 (12.5)	1 (4.17)
Moderate, *n* (%)	22 (68.75)	6 (75)	16 (66.67)
High, *n* (%)	8 (25)	1 (12.5)	1 (29.17)

Values are expressed as median [IQR] or mean ± SD. ^a^ PASI (Psoriasis Area and Severity Index) is a score that indicates the degree of involvement of psoriasis, with higher scores indicating more severe disease (PASI < 7 = mild involvement; 7 ≤ PASI ≤ 15 = moderate involvement; PASI > 15 = severe involvement). The average score was calculated for individuals with psoriasis impacting at least 3% of their body surface area at the start and who initially had a PASI score exceeding 0. ^b^ DAPSA Disease Activity Index for Psoriatic Arthritis): is a composite index indicating the degree of activity of PsA (DAPSA < 5 = remission; 5 ≤ DAPSA ≤ 14 = low activity; 15 ≤ DAPSA ≤ 28 = moderate activity; DAPSA > 28 = high activity). It is the result of the combined count of painful and swollen joints, C-reactive protein, the patient’s global assessment of arthritis measured on a VAS ranging from 0 to 100 mm, and the patient’s assessment of pain derived from the arthritis, measured on a VAS. PsA: psoriatic arthritis; CRP: C-reactive protein; csDMARD: conventional synthetic disease-modifying antirheumatic drug; ESR: erythrocyte sedimentation rate; VAS: Visual Analogue Scale.

**Table 3 diagnostics-14-00988-t003:** Evolution characteristics of 32 patients with DMARDb-refractory PsA in clinical practice who started treatment with Upadacitinib or Tofacitinib (Baseline and Month 3).

	Baseline			Month 3		
	Total (*n* = 32)	UPADACITINIB (*n* = 8)	TOFACITINIB (*n* = 24)	Total (*n* = 30)	UPADACITINIB (*n* = 8)	TOFACITINIB (*n* = 22)
Number of tender joints	5 [3–7.25]	4 ± 2.45	6.08 ± 3.59	2 [0.25–4.75] *	2.38 ± 2.45 *	2 [0.25–4.75] *
Number of swollen joints	3 [2–4]	2.38 ± 1.06	3 [2–4.25]	0 [0–2] *	1.38 ± 1.6 *	0 [0–1.75] *
CRP, mg/L	4.7 [1.48–12.9]	9.3 [2.33–24.7]	4.3 [1.3–9.7]	2.25 [1–10.1] *	2.4 [1.55–10.35]	2.15 [0.7–9.68]
ESR, mm/h	19.5 [7–28.5]	23 ± 13.09	20 [5–28.5]	13.5 [10–29.25]	12 [10–19.5]	14.5 [6.25–33]
PASI ^a^	3.7 [2.1–5.3]	2.42 ± 0.88	4.8 ± 3.23	0.8 [0–2.23] *	0 [0–0.8] *	1.58 ± 1.55 *
Mild, *n* (%)	21 (80.77)	5 (100)	16 (76.19)	24 (100)	5 (100)	19 (100)
Moderate, *n* (%)	5 (19.23)	0	5 (23.81)	0	0	0
Severe, *n* (%)	0	0	0	0	0	0
DAPSA ^b^	24.56 ± 7.97	22.57 ± 6.88	25.23 ± 8.32	13.54 ± 8.21 *	14.76 ± 7.63 *	13.10 ± 8.54 *
Remission, *n* (%)	0	0	0	1 (3.33)	0	1 (4.55)
Low, *n* (%)	2 (6.25)	1 (12.5)	1 (4.17)	17 (56.57)	4 (50)	13 (59.09)
Moderate, *n* (%)	22 (68.75)	6 (75)	16 (66.67)	10 (33.33)	4 (50)	6 (27.27)
High, *n* (%)	8 (25)	1 (12.5)	7 (29.17)	2 (6.67)	0	2 (9.09)
ACR20 ^c^ response, *n* (%)	-	-	-	20 (62.5)	6 (75)	14 (63.63)
ACR50 ^c^ response, *n* (%)	-	-	-	15 (46.88)	4 (50)	11 (50)
ACR70 ^c^ response, *n* (%)	-	-	-	8 (25)	1 (12.5)	7 (31.84)
PASI75 ^d^, *n* (%)	-	-	-	14 (58.33)	4 (80)	10 (52.63)

Values are expressed as median [IQR] or mean ± SD; * *p* < 0.05 vs. baseline (paired *t*-test or Wilcoxon signed-rank test). ^a^ PASI (Psoriasis Area and Severity Index) is a score that indicates the degree of involvement of psoriasis, with higher scores indicating more severe disease (PASI < 7 = mild involvement; 7 ≤ PASI ≤ 15 = moderate involvement; PASI > 15 = severe involvement). The calculation of the mean score was carried out only in those patients who had psoriasis involvement of at least 3% of the body surface at the beginning of the study and whose PASI score calculation was greater than 0. ^b^ DAPSA (Disease Activity Index for Psoriatic Arthritis) is a composite index indicating the degree of activity of PsA (DAPSA < 5 = remission; 5 ≤ DAPSA ≤ 14 = low activity; 15 ≤ DAPSA ≤ 28 = moderate activity; DAPSA > 28 = high activity). It is the sum of the number of painful joints, the number of swollen joints, the C-reactive protein, the patient’s global assessment of arthritis measured on a visual analog scale (VAS) ranging from 0 to 100 mm and the patient’s assessment of arthritis pain (measured on a VAS). ^c^ ACR20/50/70, according to the American College of Rheumatology criteria, is defined as a 20/50/70% reduction since the start of treatment in the number of painful joints (out of 68 joints assessed) and swollen joints (out of 66 joints assessed) and a 20/50/70% improvement in at least three of the following measures: patient and physician global assessment of arthritis (measured on a VAS), patient assessment of arthritis pain (measured on a VAS), disability (measured by HAQ-DI) or CRP level. ACR20/50/70 response rates were calculated in a total of 30 patients as two patients discontinued Tofacitinib treatment before 3 months due to mild adverse events. ^d^ PASI75: 75% or more reduction in PASI score since the start of treatment. CRP: C-reactive protein; ESR: erythrocyte sedimentation rate; VAS: Visual Analogue Scale.

## Data Availability

The data presented in this study are available on request from the corresponding author. They are not publicly available due to privacy restrictions.

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
