# Peer review of "JAKinhibs in Psoriatic Disease: Analysis of the Efficacy/Safety Profile in Daily Clinical Practice"

_diagnostics, 2024, doi:10.3390/diagnostics14100988_

Round 1

Reviewer 1 Report

Comments and Suggestions for Authors

The present paper is well-written for the most part, with an interesting topic for readers in the field. The limitations of the study are effectively presented, and the study design is on point. Please, note that is present a phrase in Spanish (paragraph 2.1) that need to be translated.

Comments on the Quality of English Language

Quality of English language is fine for the most part

Author Response

First of all, we would like to thank you for the advice and suggested corrections for this article.

After an extensive review we have performed several modifications:

  1. The order of the article sections have been updated
  2. The Abstract has been modified, reducing the number of words and depriving it of the headings.
  3. We have reviewed all the abbreviations and standardized them as requested.
  4. Duplication rate has been reduced.
  5. Although the patient cohort is small in number, with the support of experts we have carried out a basic statistical study to look for statistically significant differences in the outcome parameters previously included in the study, obtaining some interesting results.
  6. We have updated the data in the tables and figures based on the statistical results obtained.
  7. The comparison part with data from the pivotal clinical trial cohorts to highlight the differences obtained in the outcome variables despite the partially different characteristics of our patients has been included.
  8. The Review Board Statement, Informed Consent Statement, and Data Availability Statement have been included.
  9. We have performed an english grammar review of the entire article.

I hope that these modifications may be to your liking and that they have added value to this paper.

Do not hesitate to contact us again if you have any questions about the modifications made or for any explanation.

Kind regards

Reviewer 2 Report

Comments and Suggestions for Authors

The article titled "JAKinhibs in Psoriatic Disease: Analysis of the Efficacy/Safety Profile in Daily Clinical Practice" consists of 8 typical parts.
The Abstract is too long in my opinion. Additionally, please sort out the abbreviations (this note applies to the entire article). The abbreviation should be after the name, when appearing for the first time, later only abbreviation is enough. In the Abstract, we have two times PsA, PsD but we do not know what is DMARD.
It is good that the authors included a section describing the limitations of the work. But, I have a few comments. The retrospective study describes a group of 32 patients (25/8). These are indeed small groups, but this does not exclude the possibility of conducting statistical analyses. There are specially dedicated tests for small research groups. We do not know whether the presented factors changed with statistical significance or not. The “p” values are also missing from the charts. Moreover, what new does the work bring in relation to the presented clinical trials with hundreds of patients. I think it would be more interesting to show how blood parameters changed under the influence of the presented drugs. Such tests were certainly carried out during follow-up examinations. This could show the effects of the drug on liver or morphology, or perhaps on antinuclear antibodies or others.
Moreover, I believe that the comparison of the research group with clinical groups does not bring anything new:
“Comparing the baseline characteristics of the patients in our cohort with those 650
included in the pivotal RCTs [35,38], we can state that, on average, our patients were 651
younger (45.25 ± 10.71 vs 49.5 ± 12.3 [35] y 53.0 ± 12.0 años [38]), with a greater prevalence 652
of women (75% vs 49% [35] y 54% [38]) and with a shorter duration of disease (7.8 ± 5.22 653
vs 9.6 ± 7.6 [35] y 9.6 ± 8.4 [38]).”
In my opinion, the article in its presented form is not very attractive, but adding the additional results and, above all, statistical analysis would significantly increase its importance.

Author Response

(The authors gave the same response as above.)

Round 2

Reviewer 2 Report

Comments and Suggestions for Authors

Dear Authors

Thank you for including all my comments in the text. I see that a lot of work went into improving the article. Statistical analyzes undoubtedly strengthened the message of the work. In my opinion, the work can be published in its current form.